

# Spectrum analysis of inborn errors of metabolism for expanded newborn screening in Xinjiang, China

Jingying Zhu[1,2,*], Li Han[3], Pingjingwen Yang[1], Ziyi Feng[1] and Shuyuan Xue[1,2,*]

[1] Prenatal Diagnosis Center, Urumqi Maternal and Child Health Hospital, Urumqi, Xinjiang Uygur Autonomous Region, China
[2] Xinjiang Clinical Research Center for Perinatal Diseases, Urumqi, China
[3] Neonatal Department, Urumqi Maternal and Child Health Hospital, Urumqi, Xinjiang Uygur Autonomous Region, China
* These authors contributed equally to this work.

Corresponding author
Shuyuan Xue,
xueshuyuan327@gmail.com

## ABSTRACT

To determine the disease spectrum and genetic characteristics of inborn errors of metabolism (IEM) in Xinjiang province in the northwest of China, 41,690 newborn babies were screening by tandem mass spectrometry from November 2018 to December 2021. Of these, 57 newborn babies were referred for genetic analysis by next-generation sequencing, which was validated by Sanger sequencing. A total of 36 newborn babies and one relative were diagnosed with IEM, and the overall positive predictive value was 29.03%. The overall incidence of IEM in Xinjiang was 1:1,158 (36/41,690). The incidence of amino acidemias, organic acidemias, and fatty acid oxidation disorder were 1:1,668 (25/41,690), 1:4,632 (9/41,690), and 1:20,845 (2/41,690), respectively. Phenylketonuria and methylmalonic acidemia were the two most common inborn errors of metabolism (IEM), accounting for 83% (30/36) of all confirmed cases. Some hotspot mutations were observed for several IEMs, including *PAH* gene c.158G > A (p.Arg53His) and c.688G > A (p.Val230Ile) for hyperphenylalaninemia. Four mutation types of the *MMACHC* gene (*e.g.*, c.609G > A (p.Trp203Ter), c.567dupT (p.Ile190fs)) and six mutation types of the *MMUT* gene (*e.g.*, c.729_730insT (p.Asp244fs)) were found for methylmalonic acidemia. We also found 11 mutations in six genes: *PCCB, IVD, GCDH, MCCC1, SLC22A5,* and *ACADS* in this region. This study combined tandem mass spectrometry and next-generation sequencing technology for the screening and diagnosis of IEM. The study provides effective clinical guidance, and the data provide a basis for expanding newborn screening, genetic screening, and IEM gene consultation in Xinjiang, China.

## INTRODUCTION

Inborn errors of metabolism (IEMs) are a group of disorders including abnormalities in biochemical and metabolism-related enzymes, receptors, and cell membrane function, which are caused by genetic defects. IEMs include amino acidemia, organic acidemia, and

fatty acid oxidation disorders, most of which have autosomal recessive inheritance (*Ferreira & van Karnebeek, 2019*; *Gu, 2015*). The incidence of a single disease of IEM is very low, about 1/10,000 to 1/10 million, but there are many diseases (so far, more than 1,000 diseases have been found and named); therefore, the overall incidence of IEM is high and can reach 1/1,000 (*Yang, 2018*). The pathogenesis of IEM is complex, and disease onset can occur within several hours or days after birth. IEMs can cause serious harm to children and are associated with a high rate of disability and mortality, and a lack of specific clinical manifestations, and often involve missed diagnosis and misdiagnosis. The diagnosis of this kind of disease mainly depends on the laboratory testing of specific metabolic substances in the blood, urine, and other body fluids of children (*Ferreira, Hoffmann & Blau, 2019*; *Zhao & Gu, 2015*). Tandem mass spectrometry (MS/MS) can detect more than 40 indicators including amino acids and acylcarnitines at a time and can be used to screen for dozens of IEMs, with strong specificity and high sensitivity. However, the findings of MS/MS are easily affected by the children's condition, treatment, and other factors, resulting in misdiagnosis and missed diagnosis. Owing to the high genetic heterogeneity of IEM, genetic testing remains an important means of confirming disease diagnosis and guiding prognosis. As the technology becomes more mature and the cost gradually decreases, next-generation sequencing (NGS) is increasingly being used in neonatal IEM diagnosis (*Zhang et al., 2020*; *Ying & Luo, 2021*). Through gene sequencing of children with positive, suspicious samples screened by MS/MS, we can provide accurate molecular diagnoses, realize the early diagnosis and treatment of children with genetic metabolic diseases, and effectively reduce the disability rate and mortality rate of patients.

In our study, 41,690 dry blood spot samples from newborn babies were screened for IEM by MS/MS, and genetic analysis was performed in combination with NGS for positive children of screening. We aimed to explore the application effect of MS/MS combined with next-generation sequencing in the diagnosis of neonatal genetic metabolic diseases, and preliminarily obtain the incidence rate, disease spectrum, and genetic characteristics of IEMs in Xinjiang.

## MATERIALS AND METHODS

### Newborn screening

This is a retrospective study, from November 2018 to December 2021, a total of 41,690 newborn babies were referred for expanded newborn screening by MS/MS at the Neonatal Screening Center of Urumqi Maternal and Child Health Hospital in Xinjiang, China. The procedures were undertaken according to Technical Specifications for Neonatal Disease Screening (2010 Edition) (*Ministry of Health of the People's Republic of China, 2010*) in using MS/MS for newborn screening. The study protocol was reviewed and approved by the Ethics Committee of Urumqi Maternal and Child Health Hospital (Ethical approval number: XJFYLL2022003). Written informed consents were obtained from all the infants' parents.

## Instruments and reagents

American Waters Xevo TQD tandem mass spectrometer, ALLSHENG microplate constant temperature incubation oscillator, Henya ultrasonic cleaning machine, Eppendorf pipette. The nonderivatization tandem mass spectrometry screening kit (NeoBase™ Non-derivatized MS/MS Kit) and supporting reagents, including extracts and mobile phase, were produced by Perkin Elmer. The purity of the organic solvents (such as methanol) used in the experiment were all high-performance liquid chromatography (HPLC) grade. The argon gas required for the MS crash tank was purchased from a local gas company, with a purity >99.999%.

## Sample collection and delivery

According to the requirements of blood specification for neonatal genetic metabolic disease screening in the Technical Specifications for Neonatal Disease Screening (2010 Edition), sufficient post-lactation sampling should be given within 7 days after 72 h of birth. The requirements include at least three blood spots, each with a diameter of >8 mm, natural penetration of blood drops, consistent blood spots on the front and back of filter paper, no pollution, and no seepage blood ring. After natural drying, the blood tablets were stored at 2–8 °C, and the specimens were delivered to the screening center within 1 week.

## MS/MS

MS/MS was used to detect amino acids and acylcarnitines with the NeoBase™ Non-derivatized MSMS Kit. First, 3.2 mm diameter perforating forceps were used to punch the dry blood spot sample and control, and a U-shaped plate provided in the kit was used to put the corresponding blood spots in each hole. Each plate was equipped with indoor quality control samples to ensure the correctness and stability of the experiment. Second, the internal standard and daily work solution were prepared in the amino acid dry powder standard bottle and the acylcarnitine dry powder standard bottle, respectively. Each extract was 1.0 ml and was shaken at room temperature until it was completely dissolved to make an amino acid standard solution and an acyl carnitine standard solution. These were stored at 2–8 °C (they can remain stable for 30 days at this temperature). On the day of the experiment, the extract and internal standard solution were prepared as the daily working solution according to the ratio of the kit instructions. Extraction involved 100 µL of the daily working solution being added to each well using a multichannel pipette, which was sealed with sticky film. Then, the microplate was placed into a constant temperature oscillation incubator at 45 °C (700 rpm) and incubated with shaking for 45 min. The supernatant was transferred onto the V-type bottom heat-resistant plate and the whole 96-well plate was wrapped up with aluminum film. Lastly, the 96-well plate was put into the sample room and the MS detection method was used for testing.

## Genomic DNA sequencing

A kit was used to extract genomic DNA from 2 ml venous samples from the children and parents (CWE9600 Blood DNA Kit; CWBIOTECH). Qubit was used for DNA sample

quality control. Samples were stored at −20 °C for testing. DNA library preparation was undertaken, which included end repair, connector connection, and polymerase chain reaction amplification. After library preparation, quantification and dilution was completed according to the kit instructions, and then sequencing was carried out through the sequencing platform. Sequencing bioinformatics analysis included using the Burrow-Wheeler Aligner (BWA) sequence comparison method and the quality control sequence contrast human genome reference sequence. In addition, annotation software was used. Specifically, the ANNOVAR annotation mutation site was used, which is a public mutation database and associated with multiple databases, such as genome, NHLBI GO Exome Sequencing Project (ESP6500), The Single Nucleotide Polymorphism Database (dbSNP), the Exome Aggregation Consortium (EXAC), and Human Gene Mutation Database (HGMD). This resource provides information on the mutation site in the normal population frequency, sequence conservation, mutations caused by coding amino acid changes, the location in the protein structure, and mutations that are predicted to be harmful. Finally, combined with the condition of the child, the mutations were interpreted for pathogenicity according to American College of Medical Genetics and Genomics criteria and guidelines (Richards et al., 2015), and the pedigree was verified by Sanger sequencing. In this study, the next-generation sequencing and Sanger sequencing verification were sent to Shanghai Xinhua Hospital, Beijing MyGenostics, and Hangzhou Bosheng Medical Laboratory.

## Quality control

The indoor quality control products provided by the kit were used to ensure the accuracy of each experimental result. There were two kinds of indoor quality control products: low concentration value and high concentration value. Each experimental plate was equipped with two holes of high-quality control. The quality control value was recorded and used to generate an L-J quality control map: $x \pm 2SD$ was the warning limit and $x \pm 3SD$ was the runaway limit. Inter-room quality assessment of MS/MS screening of neonatal genetic metabolic diseases was organized by the Clinical Laboratory Center of the National Health Commission, and amino acid box acylcarnitine compartment quality assessment occurs every year, and all indicators were 100% qualified.

## Statistical analysis

SPSS 23.0 software was used for data collation and statistical analysis. The t-test was used for the comparison between the two groups, and the percentile method was used for the non-normal distributed data. Values were considered statistically significant at $P < 0.05$.

# RESULTS

## Performance of expanded newborn screening

From November 2018 to December 2021, a total of 41,690 newborn babies were screened by MS/MS, including 21,463 males and 20,227 females; the percentage of preterm infants was 4.60% (1,915/41,690) and that of low-birth-weight infants was 2.74% (14,141/41,690).

**Table 1 Baseline characteristics of newborn babies in the expanded newborn screening program.**

| General data grouping | Newborns without IEM | Newborns with a confirmed IEM | *P*-value |
|---|---|---|---|
| Number | 41,654 | 36 | |
| Age (mean ± SD) | 3.9 ± 6.00 | 5.86 ± 8.94 | 0.659 |
| Sex | | | |
| Male | 21,446 | 17 | 0.609 |
| Female | 20,208 | 19 | |
| Nationality | | | |
| The Han nationality | 26,248 | 26 | 0.612 |
| The Uygur ethnic group | 7,773 | 6 | |
| The Hui nationality | 3,327 | 2 | |
| Kazak | 3,276 | 1 | |
| Mongolian nationality | 432 | 1 | |
| Other | 598 | 0 | |
| Gestational age (weeks) | | | |
| <32 | 132 | 0 | 0.669 |
| 32–36 + 6 | 1,780 | 3 | |
| ≥37 | 39,736 | 33 | |
| Unknown | 6 | 0 | |
| Birth weight (g) | | | |
| <2,500 | 1,139 | 2 | 0.580 |
| 2,500–3,999 | 36,021 | 30 | |
| ≥4,000 | 4,470 | 4 | |
| Unknown | 24 | 0 | |
| Mode of delivery | | | |
| Spontaneous labor | 21,636 | 16 | 0.647 |
| Cesarean section | 19,965 | 20 | |
| Unknown | 53 | 0 | |

The median number of days for initial screening was 3 days after birth. The basic information of newborn screening is detailed in Table 1. The results of the screening of neonatal IEMs by MS/MS showed that a total of 1,457 newborn babies needed to be recalled, and the positive rate of primary screening was 3.49%, among which the number of successful recalls was 1,417 and the recall rate was 97.25%. After the recall review, the number of patients still remaining positive was 124, and the infants to be diagnosed needed further differential diagnosis and genetic testing. Finally, 36 newborn babies and one family member were diagnosed with an IEM, and the overall incidence of IEM in Xinjiang was 1:1,126 (37/41,690) (excluding one family, the overall incidence of neonatal IEM was 1:1,158). The incidence of amino acidemia, organic acidemia, and fatty acid oxidation disorder were 1:1,668 (25/41,690), 1:4,632 (9/41,690), and 1:13,896 (3/41,690), respectively (excluding one family, the incidence of abnormal neonatal fatty acid metabolism was 1:20,845). The overall positive predictive value was 29.03% (Table 2).

**Table 2 Parametric statistics of newborn IEM expanded screening.**

| Parametric statistics | Value |
|---|---|
| Total number of screening | 41,690 |
| Positive numbers of initial screening | 1,457 |
| Positive rate of initial screening (%) | 3.49 |
| Positive recall numbers of initial screening | 1,417 |
| Positive recall rate of initial screening (%) | 97.25 |
| Positive numbers after recall | 157 |
| Positive rate after recall (%) | 11.07 |
| Confirmed number | 36 |
| Positive predictive value (%) | 22.92 (36/157) |
| Incidence | 1: 1,158 |
| Amino acidemia | 1: 1,668 |
| Organic acidemia | 1: 4,632 |
| Fatty acid oxidation disorder | 1: 20,845 |

## Incidence and disease spectrum of IEM

In this study, we confirmed a total of eight different IEMs, including one amino acidemia (25 cases; 69.44%), six organic acidemia (nine cases; 25.00%), and two fatty acid oxidation disorders (two cases; 5.56%). In total, 36 newborn babies were confirmed to have an IEM (Table 3). Figure 1 showed the process of MSMS screening and diagnosing positive patients.

All the confirmed cases of amino acidemia were hyperphenylalaninemia (HPA; incidence 1/1,668). These cases including 16 newborn babies with mild hyperphenylalaninemia (the average concentration of Phe was 150.18 µmol/L and Phe/Tyr was 3.26 µmol/L). Four cases had moderate phenylketonuria (PKU) and five cases had classical PKU (the average concentration of Phe was 647.95 µmol/L and the average concentration of Phe/Tyr was 8.01 µmol/L).

The diseases diagnosed for organic acidemia included methylmalonic acidemia (MMA; five cases), propionic acidemia (PA; one case), glutaric acidemia type I (GA-I; one case), isovaleric acidaemia (IVA; one case), and three-methylcrotonyl-coenzyme A carboxylase deficiency (MCCD; one case). MMA was the most common disease for organic acidemia (incidence 1/8; 338); three cases were pure MMA and two cases were MMA combined with homocysteinemia. The mean primary screening concentrations of C3, C3/C2, and C3/C0 in all patients with MMA were 14.12, 0.47, and 0.68 µmol/L, respectively. In addition to a significant increase in C5OH and its corresponding ratio during recall and follow-up, a decrease in C0 to 2.2 µmol/L was also observed in a case with MCCD. After clinical administration of special milk powder feeding and levocarnitine supplementation, C0 returned to normal and C5OH decreased compared with before. At present, the child has good growth and development, and no abnormal clinical manifestations.

A total of two cases had fatty acid oxidation disorder (incidence 1 in 20,845), including one case of primary carnitine deficiency (PCD) and one of short-chain acyl-CoA

**Table 3 Statistics of abnormal MS/MS indicators in confirmed infants.**

| Disease | n | Incidence | Abnormal MS/MS indicators | Concentration of initial screening (µmol/L) (mean, range) | Concentration of second screening (µmol/L) (mean, range) | Reference range | Urine GC-MS result |
|---|---|---|---|---|---|---|---|
| Abnormalities in amino acid metabolism | 25 | 1/1,668 | | | | | |
| Mild hyperphenylalaninemia (HPA) | 16 | 1/2,605 | Phe | 150.18 (107.04–272.56) | 149.82 (119.60–222.09) | 20–110 | |
| | | | Phe/Tyr | 3.26 (0.79–20.75) | 1.54 (0.82–2.39) | 0.2–1.5 | |
| Phenylketonuria (PKU) | 9 | 1/4,632 | Phe | 647.95 (199.22–1,115.47) | 919.13 (387.06–1,643.87) | 20–110 | |
| | | | Phe/Tyr | 8.01 (2.41–11.41) | 14.37 (6.25–27.72) | 0.2–1.5 | |
| Abnormal metabolism of organic acids | 9 | 1/4,632 | | | | | |
| Methylmalonic acidemia (MMA) | 5 | 1/8,338 | C3 | 14.12 (6.87–22.4) | 12.37 (6.28–28.2) | 0.5–5 | Methylmalonic acid, methyl citric acid increased |
| | | | C3/C2 | 0.47 (0.26–0.71) | 1.15 (0.19–2.13) | 0–0.22 | |
| | | | C3/C0 | 0.68 (0.33–0.95) | 0.91 (0.47–1.59) | 0–0.25 | |
| Propionic acidemia (PA) | 1 | 1/41,690 | C3 | 7.96 | 14.45 | 0.5–5 | Methyl citric acid was elevated |
| | | | C3/C2 | 0.371 | 1.013 | 0–0.22 | |
| | | | C3/C0 | 0.259 | 0.356 | 0–0.25 | |
| Isovaleric acidemia (IVA) | 1 | 1/41,690 | C5 | 4.56 | 7.37 | 0.04–0.3 | Increase in isovaleryl glycine |
| | | | C5/C2 | 0.281 | 0.963 | 0–0.02 | |
| | | | C5/C3 | 2.606 | 8.57 | 0–0.3 | |
| Glutaric acidemia type I (GA- I) | 1 | 1/41,690 | C5DC + C6OH | 2.54 | 2.37 | 0.04–0.25 | Glutaric acid elevated significantly |
| | | | (C5DC + C6OH) /C8 | 63.5 | 237 | 0.5–5 | |
| | | | C0 | 15.33 | 7.19 | 9–60 | |
| 3-Methylcrotonyl-coenzyme A carboxylase deficiency (MCCD) | 1 | 1/41,690 | C4DC + C5OH | 7.5 | 12.73 | 0.08–0.4 | 3-hydroxyisovalerate, 3-methyl crotonyl glycine and methyl crotonyl glycine elevated significantly |
| | | | (C4DC + C5OH)/ C0 | 0.56 | 4.90 | 0–0.02 | |
| | | | (C4DC + C5OH)/ C8 | 375 | 1273 | 1–12 | |
| Fatty acid oxidation disorder | 2 | 1/20,845 | | | | | |
| Primary carnitine deficiency (PCD) | 1 | 1/41,690 | C0 | 5.47 | 6.04 | 9–60 | |
| Short-chain acyl-CoA dehydrogenase deficiency (SCADD) | 1 | 1/41,690 | C4 | 0.71 | 0.91 | 0.1–0.5 | Ethylmalonic acid increased |
| | | | C4/C2 | 0.06 | 0.03 | 0–0.03 | |
| | | | C4/C3 | 0.72 | 0.52 | 0.04–0.35 | |

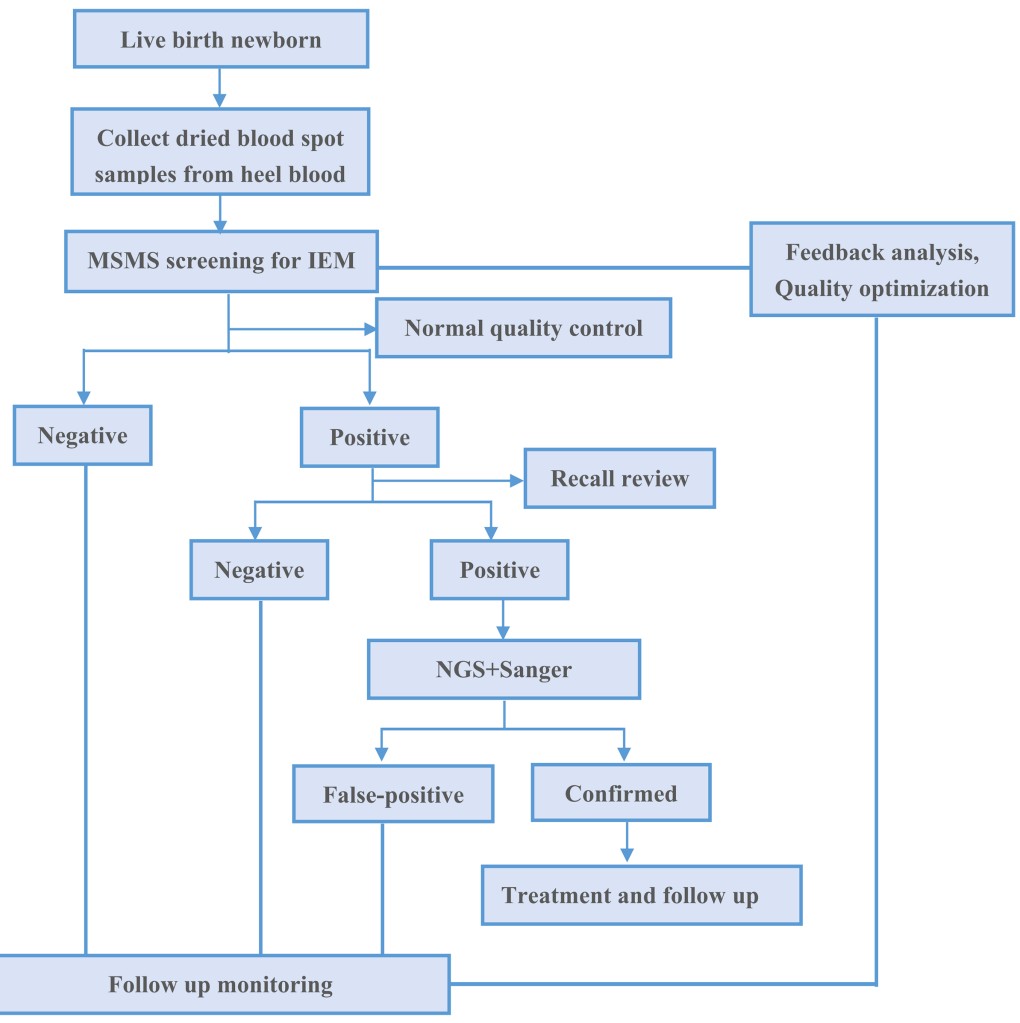

**Figure 1 MSMS screening for IEM and positive diagnosis flowchart.**

dehydrogenase deficiency (SCADD). Children with SCADD showed an increase of C4, C4/C2, and C4/C3 detected by MS/MS, and gaschromatography-mass spectrometry (GC-MS) results showed an increase in ethylmalonic acid. During the pedigree validation of the child with SCADD, the elder sister was found to carry two mutation sites in the same *ACADS* gene as this patient. The follow-up results showed that the children with SCADD and his sister had good intellectual and physical development, with normal clinical blood glucose monitoring and no abnormal clinical manifestations.

## Genetic analysis of the confirmed cases

Among the 27 newborn babies who screened positive for Phe by MS/MS, the parents of one newborn baby refused gene sequencing; another family of a newborn baby had a PKU proband, Phe > 120 μmol/L, and Phe/Tyr > 2.0 in initial screening as well as in re-examination. The case could be diagnosed through biochemical methods and special dietary treatment was performed in the early stages of screening diagnosis; therefore, NGS

**Table 4 Analysis of the gene mutation spectrum of the confirmed infants.**

| Disease | Gene (n) | Mutant site | Amino acid variant | Mutation type | Mutant regions | Pathogenicity | Allele frequency % (n) |
|---|---|---|---|---|---|---|---|
| **Amino acidemia** | | | | | | | |
| Hyperalaninemia | PAH (19) | c.158G > A | p.Arg53His | Missense | E2 | P | 23.53% (8) |
| | | c.688G > A | p.Val230Ile | Missense | E6 | P | 8.82% (3) |
| | | c.1174T > A | p.Phe392Ile | Missense | E11 | LP | 5.88% (2) |
| | | c.331C > T | p.Arg111Ter | Nonsense | E3 | P | 5.88% (2) |
| | | c.739G > C | p.Gly247Arg | Missense | E7 | LP | 5.88% (2) |
| | | c.1238G > C | p.Arg413Pro | Missense | E12 | P | 2.94% (1) |
| | | c.1256A > G | p.Gln419Arg | Missense | E12 | P | 2.94% (1) |
| | | c.1301C > A | p.Ala434Asp | Missense | E12 | P | 2.94% (1) |
| | | c.1316-1G > A | p.— | Splice acceptor | I12 | P | 2.94% (1) |
| | | c.208_210del | p.Ser70del | In-frame | E3 | P | 2.94% (1) |
| | | c.212G > A | p.Arg71His | Missense | E3 | LP | 2.94% (1) |
| | | c.311C > A | p.Ala104Asp | Missense | E3 | P | 2.94% (1) |
| | | c.355C > T | p.Pro119Ser | Missense | E4 | VUS | 2.94% (1) |
| | | c.442-1G > A | p.— | Splice acceptor | I4 | P | 2.94% (1) |
| | | c.526C > T | p.Arg176Ter | Nonsense | E6 | P | 2.94% (1) |
| | | c.707-1G > A | p.— | Splice acceptor | E6 | P | 2.94% (1) |
| | | c.722delG | p.Arg241fs | Frameshift | E7 | P | 2.94% (1) |
| | | c.728G > A | p.Arg243Gln | Missense | E7 | P | 2.94% (1) |
| | | c.740G > T | p.Gly247Val | Missense | E7 | P | 2.94% (1) |
| | | c.782G > A | p.Arg261Gln | Missense | E7 | P | 2.94% (1) |
| | | c.907delT | p.Ser303fs | Frameshift | E8 | P | 2.94% (1) |
| | | c.977G > A | p.Trp326Ter | Nonsense | E10 | P | 2.94% (1) |
| Organic acidemia (7) | | | | | | | |
| Methylmalonic acidemia | MMACHC (2) | c.445_446delTG | p.Cys149fs | Frameshift | E4 | P | 25% (1) |
| | | c.567dupT | p.Ile190fs | Frameshift | E4 | P | 25% (1) |
| | | c.609G > A | p.Trp203Ter | Nonsense | E4 | P | 25% (1) |
| | | c.217C > T | p.Arg73Ter | Nonsense | E2 | P | 25% (1) |
| | MMUT (3) | c.914T > C | p.Leu305Ser | Missense | E5 | P | 16.7% (1) |
| | | c.1106G > A | p.Arg369His | Missense | E6 | P | 16.7% (1) |
| | | c.1677-1G > A | p.- | Splice acceptor | I9 | P | 16.7% (1) |
| | | c.729_730insTT | p.Asp244fs | Frameshift | E3 | P | 16.7% (1) |
| | | c.1663G > A | p.Ala555Thr | Missense | E9 | LP | 16.7% (1) |
| | | c.632_636del | p.Glu211fs | Frameshift | E3 | P | 16.7% (1) |
| Propionic acidemia | PCCB (1) | c.1316A > G | p.Tyr439Cys | Missense | E13 | P | 50% (1) |
| | | c.1364G > T | p.Trp455Leu | Missense | E13 | LP | 50% (1) |
| Isovaleric acidemia | IVD (1) | c.1195G > C | p.Asp399His | Missense | E12 | LP | 100% (1) |

(Continued)

| | Table 4 (continued) | | | | | | | |
| --- | --- | --- | --- | --- | --- | --- | --- |
| Disease | Gene (n) | Mutant site | Amino acid variant | Mutation type | Mutant regions | Pathogenicity | Allele frequency % (n) |
| Glutaraemia type I | GCDH (1) | c.532G > A | p.Gly178Arg | Missense | E7 | P | 50% (1) |
| | | c.892G > A | p.Ala298Thr | Missense | E9 | LP | 50% (1) |
| 3-Methylcrotol glycine carboxylase deficiency | MCCC1 (1) | c.1977G > A | p.Lys659Lys | Missense | E18 | LP | 50% (1) |
| | | c.1679dupA | p.Asn560fs | Frameshift | E17 | P | 50% (1) |
| **Fatty acid oxidation disorder (2)** | | | | | | | |
| Primary carnitine deficiency | SLC22A5 (1) | c.254_264dupGGCTCGCCACC | p.Ile89Glyfs*45 | Frameshift | E1 | P | 50% (1) |
| | | c.629A > G | p.Asn210Ser | Missense | E3 | P | 50% (1) |
| Short-chain acyl-CoA dehydrogenase deficiency | ACADS (2) | c.989G > A | p.Arg330His | Missense | E8 | VUS | 50% (1) |
| | | c.164C > T | p.Pro55Leu | Missense | E2 | LP | 50% (1) |

gene sequencing was not performed. The remaining 25 screen-positive infants were tested by NGS sequencing, which detected a total of 22 different mutation sites of the PAH gene: c.158G > A was the most common, followed by c.688G > A, c.331C > T, c.1174T > A, as well as c.739G > A. The detailed gene mutations are shown in Table 4, no novel mutations were found in our study. According to the clinical manifestations and final diagnosis of disease, it was found that the infants with mutations in the locus c.158G > A, c.688G > A, c.331C > T, and c.1174T > A of the PAH gene showed mild to moderate hyperphenylalaninemia clinically.

Genetic analysis of the 11 infants with non-PKU IEMs showed that 91% (43/47) carried two mutated genes associated with specific disease. A total of 21 mutations were detected in eight genes associated with IEM, including 12 missense mutations, six frameshift mutations, two nonsense mutations, and one splice site mutation. According to the American College of Medical Genetics and Genomics classification criteria, 66.67% of variants (14/21) mutations were pathogenic, 28.57% of variants (6/21) mutations were likely pathogenic, and 4.76% of variants (1/21) mutations were of unknown significance.

Among the abnormalities in organic acid metabolism, five newborn babies were diagnosed with methylmalonic acidemia. Two of these cases were methylmalonic acidemia combined with homocysteinemia, carrying four types of mutations in the *MMACHC* gene, and studies have shown that c.609G > A may be the hot spot mutation carried in local patients with concurrent methylmalonic acidemia (*Zhang et al., 2021*). This mutation was also found in our study, but whether it is a hotspot mutation in this region cannot be determined and needs further verification with the accumulation of cases and data. Three cases were isolated methylmalonic acidemia, and carried six mutation types in the *MMUT* gene. In one case with confirmed isovaleric acidemia, only a possible pathogenic mutation site of the *IVD* gene was found, and the final diagnosis was made based on blood MS/MS values, combined with urinary GC-MS analysis and clinical manifestations. A patient with

SCADD was confirmed by positive MS/MS for neonatal screening (persistently high C4 and C4/C3 levels in blood). Combining urine organic acid testing (increased urine ethylmalonic acid) and gene sequencing, we found that the elder sister also had the same *ACADS* mutation site c.989G > A (paternal source) and c.164C > T (maternal source). During clinical follow-up, neither the child nor his sister had clinical manifestations such as feeding difficulties, hypoglycemia, developmental delay, hypotonia, and epilepsy. The children also had good intellectual development, which is consistent with previous studies (*Wang et al., 2020*), as SCADD mostly does not have special clinical manifestations.

## DISCUSSION

In China, as a result of the uneven development of science and technology, the application of MS/MS technology in the screening of IEMs has regional differences. The epidemiology, technical performance, and clinical effectiveness of MS/MS screening has been studied in many provinces and cities in China. The results have shown that the combination of NGS and MS/MS is an intensive program of IEM screening, and the IEM disease spectrum and genetic characteristics vary greatly in different regions. This study is the first related report on the incidence, disease spectrum, and genetic characteristics of IEM in a population in Xinjiang, China.

In our study over 3 years, a total of 41,690 newborn babies were screened for IEM and, finally, 36 newborn babies and one relative received a diagnosis of IEM. The overall incidence of IEM in Xinjiang was 1:1,158, which is higher than the overall incidence previously reported by the Chinese mainland (1:3,300) (*Gu, 2014*). Moreover, compared with the incidence of IEM in different regions of China, it is similar to Xi'an in northern China (1:1,898) (*Zhang et al., 2021*) and Jining city (1:1,941) (*Yang et al., 2020b*), and higher than that in Jiangsu province (1:2,663) (*Yang et al., 2019*), Changsha city (1:4,237) (*Li et al., 2022*), and Zhejiang province (1:5,626) (*Huang et al., 2012*). This finding indicates that IEM is not uncommon in Xinjiang, China; however, in the face of no large-scale newborn expended screening in this region, it is necessary to consider including MS/MS in the newborn screening system in Xinjiang. By analyzing the type of gene mutations in the confirmed cases, the *PAH* gene c.158G > A (p.Arg53His) and c.688G > A (p.Val230Ile) point mutation are the most common, it is similar to Fujian Province in Southeastern China (*Zhou et al., 2022*; *Santos et al., 2008*), but it is significantly different from Japan (*Okano et al., 2011*) and Brazil (*Steven et al., 2007*) (the most common mutations arec.1238G > C (p.Arg413Pro) and c.1162G > A (p.Val388Met), respectively), and it is also different from Europe and America such as Germany (*Johamnes, 2003*) and the United States (*Steven et al., 2007*), which the most common mutation is c.1222A > T ( p. Arg408Try). Four mutation types of the *MMACHC* genes were found in MMA cases in this region (including c.445_446del TG, c.567dup T, c.609G > A and c.217C > T). Six mutation types of the *MMUT* gene were found (including c.914T > C, c.1106G > A, c.1677-1G > A, c.729_730ins T, c.1663G > A, c.632_636del), and 11 mutation types were found for six genes: *PCCB, IVD, GCDH, MCCC1, SLC22A5*, and *ACADS* (Table 4).

Despite the significant differences in the disease spectrum of IEM in different regions, the widely reported PKU is the most common amino acidemia. Moreover, numerous

studies indicate that PKU is more common in northern China than in southern China, and our study findings confirm these conclusions. Genetic analysis indicated that the most common mutation in the *PAH* gene was c.158G > A, and the patient carrying this pathogenic gene showed a mild elevation in phenylalanine, which is consistent with previous reports (*Wang et al., 2021*; *Zhang et al., 2022*) Such patients do not need special dietary treatment and have a good prognosis.

Data obtained from various countries worldwide suggest that the estimated incidence of MMA ranges from 1:48,000 to 1:250,000 (*Weisfeld-Adams et al., 2013*). The reported incidence of MMA ranges from 1:115,000 in Italy to 1:169,000 in Germany (*Melo et al., 2011*). In our Study, MMA was the second most common IEM in Xinjiang and also showed the highest incidence of organic acidemia (1:8,338). This incidence was lower than that in Xi'an city (1:6,960) (*Zhang et al., 2021*), Henan province (1:6,032) (*Zhao et al., 2016*), and Jining city (1:5,590) (*Yang et al., 2020b*), but significantly higher than that in Jiangsu province (1:35,734) (*Yang et al., 2019*) and Zhejiang province (1:46,500) (*Hong et al., 2017*). Therefore, our study further demonstrates that the incidence of MMA in northern China is higher than that in southern China. Most reports show that MMA with homocysteinemia caused by *MMACHC* gene mutation is a common type of MMA in mainland China. However, in this study, five patients were diagnosed with MMA, and 60% had isolated MMA, whereas 40% had MMA with homocysteinemia, which is similar to the situation in Xi'an (*Zhang et al., 2021*). This difference could relate to the small number of confirmed cases in this study and the influence of geographical differences. Studies have shown that two common mutation of the *MMACHC* gene are c.609G > A and c.567dup T (*Zhang et al., 2022*; *Zhou et al., 2019*), and these two kinds of *MMACHC* mutations also appeared in this study. The cblc type is the most common in MMA, and the cblc incidence is 1:32,271 in Italy (*Ruoppolo et al., 2022*), and 1:85,000 in Portugal (*Nogueira et al., 2017*), and 1:100,000 in Spain (*Pajares et al., 2021*). By contrast, a pilot study in Beijing, China had the highest incidence of 1:11,730 (*Yang et al., 2020a*). The presence of elevated MMA and homocysteine levels suggests an inborn error of cobalamin metabolism in clbC patients. Two cases of cblc type MMA were confirmed in this study, ater treatment with hydroxocobalamin, levocarnitine and betaine, C3 concentration level and the condition are under the control, but our results have not yet defined either genotype-phenotype correlation, and the incidence of this disease requires further data analysis. Studies have shown that c.729_730 insT is the most common MMUT mutation in China (*Zhang et al., 2022*; *Kang et al., 2020*), and this point mutation was also found in this study. Three isolated MMA cases with special formula diet treatment still showed delayed growth and development. The molecular characteristics suggest the need for accurate and convenient genetic counseling for MMA patients and early prenatal diagnosis for high-risk familiess (*Hu et al., 2018*). It is critical to increase the coverage of newborn screening to allow early detection and treatment of MMA and combined MMA and homocystinuria patients.

In a confirmed patient with PA, the *PCCB* gene c.1316 A > G and c.1364G > T mutations were found. Despite timely treatment, the patient was admitted multiple times because of metabolic acidosis and exhibited convulsions and delayed growth and development. In patients with PA and MMA (including *MMAHC*), the clinical course

usually begins with acute metabolic decompensation in the neonatal period, often leading to irreversible neurological damage (*Haijes et al., 2020*). One IVA patient also showed delayed growth and development. A patient with GA-I, despite timely treatment with a special formula diet and levocarnitine, still experienced clinical manifestations such as metabolic acidosis and delayed growth and development, and unfortunately died at one year old. However, in our case, similar to the case described by *Martín-Rivada et al. (2022)*, most patients are symptomatic before the outcome of newborn screening. It is worth noting that MS/MS screening in Xinjiang started late, and with the continuous accumulation of data, more objective and reliable information about the spectrum of disease in MMA will be obtained. With respect to genotype-phenotype correlations, the limited number of patients included in our study, frequency of compound heterozygotes and lack of enzymatic detections made it difficult to assess the relationship between gene mutations and clinical manifestation.

Although a patient with glutaraldehyde type I was treated early and received a special milk powder diet, the clinical manifestation was delayed growth and development, and the child died at the age of 1 year. In a isovaleremia case, only one site of a possible pathogenic mutation in the *IVD* gene was identified (c.1195G > C), and the final diagnosis was made based on the blood MS/MS value and clinical manifestations. In the above case, biochemical screening was still irreplaceable. Therefore, gene sequencing cannot replace MS/MS screening, and considering the false-negative results of MS/MS screening, neonatal metabolic and genetic screening analysis must be combined to help improve the diagnostic rate of genetic metabolic diseases.

In our study, fatty acid oxidation disorders represented only 5.56% of all confirmed cases, including PCD and SCADD. Owing to the heterogeneous and nonspecific clinical manifestations of primary carnitine deficiency, misdiagnoses and missed diagnoses are common and some patients can have no abnormal manifestations for life, whereas for other patients it is potentially fatal without treatment. A patient with primary carnitine deficiency who underwent treatment with levocarnitine showed good growth and development. None of the confirmed newborn babies with SCADD (or the relative) showed clinical symptoms during the follow-up period and they had normal physical and intellectual development. This is consistent with previous studies, in which SCADD is mostly benign clinically. Dietary management is mainly conducted after early diagnosis to reduce the possibility of SCADD and avoid long fasting and hypoglycemia. The treatment during acute onset is similar to that of other fatty acid metabolism disorders (*Wang et al., 2020*; *Gong et al., 2022*). Early diagnosis by expanding newborn screening allows effective intervention before the onset of clinical symptoms, thereby reducing the risk of poor prognosis. However, owing to the insufficient number of cases, further cases need to be accumulated to obtain the hot spot mutations in the region.

The use of MS/MS combined with NGS can play a prominent role in the detection of variants involving multiple subtypes of genes. It can also be used to contribute to the active detection and evaluation of high-risk children with high clinical heterogeneity and to investigate the phenotype of complex metabolic diseases in the neonatal period, so as to provide positive therapeutic measures. For example, MS/MS indexes C3 and C4, which

correspond to a variety of diseases, and NGS technology can be used for differential diagnosis to identify disease subtypes, and provide correct and effective treatment plans for clinical practice. However, we concede that our results may not accurately reflect the prevalence of IEM or the incidence of IEMs and disease spectrum of IEMs in Xinjiang because of the lack of extensive information on expanding newborn screening.

## CONCLUSIONS

We present the extended newborn screening results with MS/MS in a population in Xinjiang, China. The prevalence, disease spectrum, and genetic characteristics of confirmed IEM were preliminarily clarified. Newborn IEM are not rare in Xinjiang, China, and PKU and MMA are the most common IEM types in this region. This screening information provides effective clinical guidance for the promotion and application of novel screening MS/MS combined with genetic analysis for the diagnosis of IEM. In addition, the hot spot mutation genes we identified may be potential candidates for genetic screening, which will provide insights for genetic counseling and genetic diagnosis of IEM. We still need to accumulate more clinical data to standardize and optimize MS/MS to expand newborn screening. Combining MS/MS and genetic screening is a new model for future screening of genetic metabolic diseases in newborn babies.

## ACKNOWLEDGEMENTS

Throughout the writing of this dissertation I have received a great deal of support and assistance. I would first like to thank my director, Shuyuan Xue, whose expertise was invaluable in formulating the research questions and methodology. Your insightful feedback pushed me to sharpen my thinking and brought my work to a higher level. I would particularly like to acknowledge my teammate, for their wonderful collaboration and patient support. Finally, we thank the editor and the reviewers for their useful feedback that improved this article.

### Funding

This work was supported by the Natural Science Foundation of Xinjiang Uygur Autonomous Region, China (No. 2021D01B16) and the Science and Technology Innovation Team (Tianshan Innovation Team) Program (No. 2022TSYCTD0016). The funders had no role in study design, data collection and analysis, decision to publish, or preparation of the manuscript.

### Grant Disclosures

The following grant information was disclosed by the authors:
Natural Science Foundation of Xinjiang Uygur Autonomous Region, China: 2021D01B16.
Science and Technology Innovation Team (Tianshan Innovation Team) Program: 2022TSYCTD0016.

## Competing Interests

The authors declare that they have no competing interests.

## Author Contributions

- Jingying Zhu conceived and designed the experiments, performed the experiments, analyzed the data, prepared figures and/or tables, authored or reviewed drafts of the article, and approved the final draft.
- Li Han conceived and designed the experiments, analyzed the data, authored or reviewed drafts of the article, and approved the final draft.
- Pingjingwen Yang performed the experiments, prepared figures and/or tables, and approved the final draft.
- Ziyi Feng performed the experiments, prepared figures and/or tables, and approved the final draft.
- Shuyuan Xue conceived and designed the experiments, analyzed the data, authored or reviewed drafts of the article, and approved the final draft.

## Human Ethics

The following information was supplied relating to ethical approvals (*i.e.*, approving body and any reference numbers):

The Ethics Committee of Urumqi Maternal and Child Health Hospital approval to carry out the study (Ethical approval number: XJFYLL2022003).

## Data Availability

The raw data are available in the Supplemental File. The sequences are available at Genbank: PRJNA1181303, SRR31252037, SAMN44553741.

## Supplemental Information

Supplemental information for this article can be found online at http://dx.doi.org/10.7717/peerj.18173#supplemental-information.

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
