# Peer review of "Spectrum analysis of inborn errors of metabolism for expanded newborn screening in Xinjiang, China"

_PeerJ, doi:10.7717/peerj.18173_

## Round 0.1 · original submission · Major Revisions

The reviewers found your manuscript interesting, however they had a number of concerns that need to be addressed. First, they would like you to include data on population differences from around the world for the most common PAH mutations, as well as on the incidence of MMA from other regions of the world. Additionally, information on the specific tool you utilized to determine the pathogenicity of variants reported in table 4 needs to be included. The reviewers also suggest that you provide more detail on how positive cases were evaluated and whether any genetic evaluation was performed on the false positives. Lastly, they requested that you include a flow-chart summarizing your study (number of newborns screened, positives identified, recall and follow-up numbers etc), as well as a table showing the false positive rates, recall and additional investigations performed to rule out the specific conditions.

Please, submit a detailed rebuttal which shows where and how you have taken all comments and suggestions into consideration. If you do not agree with some of the reviewers’ comments or suggestions, please explain why. Your rebuttal will be critical in making a final decision on your manuscript. Please, note also that your revised version may enter a new round of review by the same or by different reviewers. Therefore, I cannot guarantee that your manuscript will eventually be accepted.


**Language Note:** PeerJ staff have identified that the English language needs to be improved. When you prepare your next revision, please either (i) have a colleague who is proficient in English and familiar with the subject matter review your manuscript, or (ii) contact a professional editing service to review your manuscript. PeerJ can provide language editing services - you can contact us at [email protected] for pricing (be sure to provide your manuscript number and title). – PeerJ Staff

Reviewer 1 ·

Basic reporting

Please see the additional comments section.

Experimental design

Please see the additional comments section.

Validity of the findings

Overall, while I find this paper interesting, I would appreciate a more comprehensive analysis of the results by incorporating data from various global regions could provide a useful comparison for readers.

Additional comments

Dr. Zhu and colleagues have provided a description of the findings on 41690 newborns from Xinjiang. This work is, according to the authors, the first description of the IEM in Xinjiang. While I find the paper interesting, I would like to encourage the authors to take into account the following recommendations:

1. Abstract: What does the “excluding one family” means? Why was a family excluded?
2. Abstract and along the text: please provide the amino acid substitutions (p.) for the mutations and not only the c. of missense replacements ´
3. Abstract: please replace “mutation types of six genes” to “mutations in six genes”
4. Material and Methods, line 115: please provide the name of the kit used for DNA extraction
5. Material and Methods, line 123, Replace “Ann-novar” by “ANNOVAR” and provide the corresponding reference (PMID 20601685).
6. Discussion, line 256: replace “mutation is the most common” to “mutations are the most common”
7. Discussion, line 258: Why is the c.729_730insT mutation highlighted in the text?
8. Discussion, lines 261-265: please provide references to support this statement and add the incidence of PAH and compare it with previous reports on Chinese regions and in other regions of the world.
9. Discussion, line 264: Is the Arg53His mutation in PAH the most frequent mutation in all populations around the world? Can the authors discuss the population differences related to the most frequent alleles in PAH?
10. Discussion, lines 267-272: Please add data on the incidence of MMA in other regions of the world
11. Discussion, line 285: replace “c.729_730 ins T” to “c.729_730insT”
12. Discussion, lines 325-334: The message in this text is not clear to me. In my opinion the text should be rephrased and simplified for clarity. For example, the sentence “To control the occurrence of false negatives, we strictly set cut-off values, which simultaneously led to a high initial screening positive rate, bringing a psychological and financial burden for newborn guardians.” is completely unclear.
13. Lines 348-356: I encourage the authors to revisit the acknowledgements, as they appear to have been originally written for a dissertation.
14. Table 2: “Incidence” indicates total Incidence?
15. Table 4: The Mutant regions column should be revised as the c.158G>A does not belong to E1
16. Table 4: Which tool was used for the data included in the Pathogenicity column? I could not find any reference to any pathogenicity assessment tool in the material and methods section.
17. Are any of the mutations on Table 4 novel ones (not previously reported in the literature)? A statement about it should be included in the text and in the abstract.
18. The Supplemental files are not mentioned in the main text.

·

Basic reporting

No comment

Experimental design

No comment

Validity of the findings

External and internal validity- appropriate

Additional comments

Good effort; however there are a few concerns-
1. The mean number of days for initial screening was 3.90 ± 6.00 days after birth.- Statistically, this is inaccurate; should be expressed as median and IQR as this is a non-normal distribution. Same comment applicable for the table.
2. How were the positive cases evaluated? For example- C3 elevation-B12 estimation conducted? Any statistically significant difference between the values
true positives and false positives.
3. Suggest including a table on False positive rate, recall, and the investigations performed to rule out the actual condition
4. Any genetic evaluation/screening performed on biochemical samples that were positive (not the true positives). If not done, it is difficult to ascertain the role of genetics in newborn screening, the conclusion could be that it is performed as confirmatory.
5. "Genetic analysis of the 10 infants with non-PKU IEMs showed that 91% (43/47) carried two mutated genes associated with a specific disease."- This sentence is not clear- it should be 11 infants with non-PKU isn't it?
6. Clinical information of certain disorders are provided, uniformly all the positives should have the clinical descriptions.
7. To include a flow chart- depicting the flow of patients and the positives.

---

## Round 0.2 · Minor Revisions

Thank you for thoroughly addressing the reviewers' comments which resulted in a great improvement in the quality of your manuscript. One of the original reviewers was able to review the revised manuscript and has a few minor suggestions with regard to finalizing your manuscript.

Reviewer 1 ·

Basic reporting

no comment

Experimental design

no comment

Validity of the findings

no comment

Additional comments

The authors have addressed my comments but there are few minor corrections still needed:

1. Figure 1- Please correct “Negtive”, and complete the words “False-positi” and “Treatment and”
2. Please remove the spaces at the genes name (e.g. MCCC 1 shall be replaced by MCCC1).
3. Line 264: “2007) [10],which the most common mutation is c.1222A>T( p.Arg408TRY)” shall be replaced by “2007), in which the most common mutation is c.1222A>T ( p.Arg408Try)”
4. There are cases where unnecessary spaces are found between words, and lack of space between words in other cases, therefore I would like to ask authors to correct this in the revised version.

---

## Round 0.3 · accepted · Accept

Thank you for addressing the reviewers' comments and thus greatly improving your manuscript.

Reviewer 1 ·

Basic reporting

I thank the authors for addressing my comments and suggestions. I have a minor correction:

1. Line 264: Please replace “the most common mutation is c.1222A>T ( p.Arg408Try)” to “the most common mutation is c.1222A>T ( p.Arg408Tyr)”

Experimental design

no comment

Validity of the findings

no comment